# Efficient and Context-Aware Label Propagation for Zero-/Few-Shot Training-Free Adaptation of Vision-Language Model

**Yushu Li**[1,2,4*]**, Yongyi Su**[1,2*]**, Adam Goodge**[2]**, Kui Jia**[3]**, Xun Xu**[2†]
[1] South China University of Technology
[2] Institute for Infocomm Research (I²R), A*STAR
[3] School of Data Science, The Chinese University of Hong Kong, Shenzhen
[4] Shanghai AI Laboratory
`eeyushuli@mail.scut.edu.cn, eesuyongyi@mail.scut.edu.cn,`
`goodge_adam_david@i2r.a-star.edu.sg, kuijia@cuhk.edu.cn,`
`xu_xun@i2r.a-star.edu.sg`

## Abstract

Vision-language models (VLMs) have revolutionized machine learning by leveraging large pre-trained models to tackle various downstream tasks. Although label, training, and data efficiency have improved, many state-of-the-art VLMs still require task-specific hyperparameter tuning and fail to fully exploit test samples. To overcome these challenges, we propose a graph-based approach for label-efficient adaptation and inference. Our method dynamically constructs a graph over text prompts, few-shot examples, and test samples, using label propagation for inference without task-specific tuning. Unlike existing zero-shot label propagation techniques, our approach requires no additional unlabeled support set and effectively leverages the test sample manifold through dynamic graph expansion. We further introduce a context-aware feature re-weighting mechanism to improve task adaptation accuracy. Additionally, our method supports efficient graph expansion, enabling real-time inductive inference. Extensive evaluations on downstream tasks, such as fine-grained categorization and out-of-distribution generalization, demonstrate the effectiveness of our approach. The source code is available at `https://github.com/Yushu-Li/ECALP`.

## 1 Introduction

Emerging foundation models have transformed the traditional paradigm of machine learning model development (Jia et al., 2021; Kim et al., 2021; Li et al., 2022). By pre-training vision-language models (VLMs) on image-language pairs at a massive scale, these models have demonstrated strong inference capabilities across a wide range of downstream tasks, all while reducing the need for extensive data collection and labeling (Zhou et al., 2022b; Zhang et al., 2022; Shu et al., 2022).

Recent research on adapting pre-trained VLMs to downstream tasks has primarily focused on improving label efficiency, training efficiency, and data efficiency. DMN (Zhang et al., 2024b) represents a comprehensive approach that addresses all three dimensions. Specifically, DMN leverages text prompts, test samples, and few-shot samples to construct a three-branch classifier, with final predictions being the fusion of the individual branches. While DMN achieves state-of-the-art performance on fine-grained categorization tasks, it requires tuning a task-specific coefficient to fuse predictions, often using the test set for hyperparameter tuning in the absence of a dedicated validation set. Additionally, DMN introduces a memory bank of test samples to synthesize an adaptive classifier for each sample. However, we hypothesize that these test samples can be used more effectively—not just for classifier synthesis but also to better capture the data manifold for transductive inference (Joachims, 2003), particularly when labeled samples are limited.

---

*Equal contribution. †Correspondence to Xun Xu: xu_xun@i2r.a-star.edu.sg. This work was done during Yushu Li and Yongyi Su's visit to I²R.

To reduce the need for task-specific hyperparameter tuning and better utilize test samples, we propose a graph-based adaptation and inference method for downstream tasks. Our approach dynamically constructs a graph based on available few-shot samples (if applicable), test samples, and text prompts linked to semantic labels. This graph captures the intrinsic data manifold, and we use label propagation (Zhu & Ghahramani, 2002) for inference. Unlike DMN, this method eliminates the need for task-specific hyperparameter tuning and enhances the use of information embedded in the unlabeled test samples.

ZLaP (Kalantidis et al., 2024) recently introduced a zero-shot VLM adaptation method based on label propagation, employing an external dataset to build the manifold and using a closed-form solution to propagate labels from text prompts to test samples. However, we argue that ZLaP is suboptimal for label-efficient VLM adaptation for three key reasons. First, a closed-form solution becomes computationally expensive for larger datasets like ImageNet, which includes 50,000 test samples, due to the costly inversion of the Laplacian matrix, limiting the graph's connectivity. Second, using a static graph based solely on the training set without incorporating test samples fails to leverage the test data manifold, which can lead to performance degradation when there is a distribution shift between the training and test sets. Third, relying on cosine similarity to measure affinities between test samples and prompts can be problematic. Since VLMs are pre-trained on diverse image-text pairs, their vision-encoded features may capture irrelevant semantic information, such as background objects or image style, making cosine similarity biased for downstream tasks.

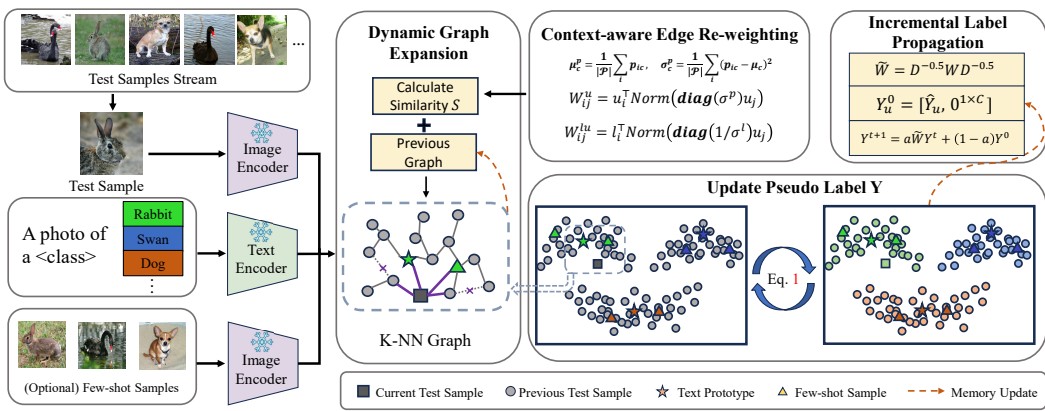

Figure 1: Illustration of the overall framework of **ECALP**. We use the text prompts, few-shot samples, and test samples to build the graph. The graph is dynamically expanded upon seeing a new sample with context-aware re-weighted edge weights. An iterative solution is adopted to predict labels for all test samples.

In this work, we introduce a holistic label propagation approach to label-efficient adaptation of VLMs with an overview presented in Fig. 1. To address the challenge of computational efficiency, we employ an iterative solution to label propagation instead of the closed-form solution, which could benefit from incremental label propagation and label reset. Furthermore, we employ a context-aware feature dimension re-weighting to better adapt to downstream tasks. We require only the text prompts and/or few-shot samples to provide contextual information for feature re-weighting. We also present an efficient graph expansion mechanism to allow inductive inference on a stream of test samples, without requiring all test data for transductive labeling. Our proposed method is referred to as Efficient and Context-Aware Label Propagation (**ECALP**). Experiments are conducted on a wide range of downstream tasks, including fine-grained categorization, distribution shift and few-shot categorization. We summarize the contributions of this work as follows:

- We propose a unified label-efficient adaptation of vision-language models from a label propagation perspective. Compared with state-of-the-art methods, the proposed method achieves higher inference speed and fixed hyperparameters.

- We account for the diversity of information captured by the vision encoder and propose to re-weight the vision feature dimensions using the statistics of the text embeddings and/or few-shot samples' features from the downstream task.

- We demonstrate the effectiveness of our approach through experiments on a wide range of downstream tasks, including fine-grained categorization and out-of-distribution generalization, achieving state-of-the-art results.

## 2 RELATED WORK

### 2.1 ZERO/FEW-SHOT ADAPTATION OF VISION-LANGUAGE MODEL

Vision-language models (VLMs) (Zhang et al., 2024a; Jia et al., 2021; Li et al., 2022; 2023) have achieved remarkable success in open-vocabulary classification tasks. However, their performance significantly declines in fine-grained scenarios (Zhao et al., 2017) or under distribution shifts (Hendrycks & Dietterich, 2019), highlighting the limitations of large-scale pre-training. To address these challenges, two primary strategies have been developed to improve VLM adaptability in zero-shot and few-shot settings: prompt tuning and adapter tuning.

Prompt tuning (Yoon et al., 2024; Feng et al., 2023), inspired by techniques in language models (Brown et al., 2020), improves performance by optimizing learnable input prompts. For instance, CoOp (Zhou et al., 2022b) and CoCoOp (Zhou et al., 2022a) replace static text prompts with learnable word vectors, fine-tuned via few-shot classification loss. TPT (Shu et al., 2022) and Swap-Prompt (Ma et al., 2024) extend this concept to zero-shot settings, employing contrastive learning on augmented image inputs to generate pseudo-labels. Although effective, prompt tuning requires full backpropagation through the computational graph, which results in high computational costs.

Adapter tuning (Zhu et al., 2023; Wang et al., 2024), exemplified by Clip-Adapter (Gao et al., 2024), adapts image or text features using lightweight adapters, thereby avoiding full backpropagation through complex encoders. Recently, training-free adapter approaches, such as Tip-Adapter (Zhang et al., 2022), have gained traction by leveraging few-shot image features as prototypes, removing the need for learned adapter weights. DMN (Zhang et al., 2024b) and TDA (Karmanov et al., 2024) extend these methods to zero-shot scenarios by building memory banks from test data and pseudo-labels. However, these approaches often introduce additional costs due to image augmentation and repeated sample processing. Moreover, their reliance on complex architectures and extensive hyper-parameter tuning limits their practicality in real-world applications.

In this work, we propose a unified, training-free adaptation framework for VLMs based on label propagation. Our method exploits the manifold structure of test data, eliminating the need for augmentation and exhaustive hyperparameter searches, offering a more efficient and practical solution compared to previous approaches.

### 2.2 LABEL PROPAGATION

Label propagation (LP) is a well-established method widely used for label-efficient learning (Zhu et al., 2003; Iscen et al., 2019; Xu & Lee, 2020; Zhu & Koniusz, 2023). LP operates under the assumption that labels vary smoothly across a graph, with adjacent nodes likely sharing the same label. In transductive learning, where all test samples are visible during inference, LP propagates labels from labeled nodes to unlabeled ones. It has been particularly effective in retrieval tasks, where a new query is appended to the graph, and its affinity is propagated to prototype nodes (Iscen et al., 2017). Recent studies have demonstrated the benefits of LP in adapting VLMs to downstream tasks (Hu et al., 2024; Kalantidis et al., 2024), where a graph is constructed from downstream test samples, and labels are propagated from text prompt prototypes to test samples using a closed-form LP solution. Despite its strengths, this approach faces several challenges, as previously discussed. In response, we propose a holistic solution for label-efficient adaptation of VLMs, built upon a label propagation framework.

## 3 METHODOLOGY

### 3.1 PROBLEM FORMULATION

We begin by formally defining the task of zero-/few-shot adaptation of vision-language models. We denote the image encoder and text encoder of a VLM, such as CLIP, as $f(x)$ and $g(z)$

respectively. The downstream task consists of the encoded features of unlabeled testing data $\mathcal{D}_u = \{u_i = f(x_i)\}_{i=1\cdots N_u}$ and optionally the features and labels of few-shot labeled data $\mathcal{D}_l = \{l_j = f(x_j), y_j\}_{i=1\cdots N_l}$. Known class names are combined with prompt templates to form textual prompts. Each class consists of multiple textual prompts $\{g(z_{cm})\}$ and we take the average as the textual prototype $p_c = \frac{1}{M}\sum_m g(z_{cm})$ for the $c$-th class and denote all prototypes as $\mathcal{P} = \{p_c\}_{c=1\cdots C}$. The task is to infer the labels of the unlabeled data $\mathcal{D}_u$ using the textual prototypes $\mathcal{P}$ and optionally the few-shot labeled data $\mathcal{D}_l$.

## 3.2 REVISITING TRANSDUCTIVE LABEL PROPAGATION

We revisit transductive label propagation and introduce the graph construction process for a holistic, label-efficient approach to VLM adaptation. Instead of directly comparing the similarity between unlabeled samples and class prototypes, we aim to exploit the manifold of all available downstream data. We denote a graph built upon all downstream data as $G = (\mathcal{V}, \mathcal{E})$ where each node is a downstream data sample or prototype, i.e. $v_i \in \mathcal{P} \cup \mathcal{D}_l \cup \mathcal{D}_u$. The adjacency matrix for the graph is denoted as $W \in \mathbb{R}^{N_p + N_l + N_u}$. A normalized adjacency matrix is obtained as $\tilde{W} = D^{-0.5}WD^{-0.5}$ with $D$ being the degree matrix. Labels $Y$ are propagated and refined according to the rule in Eq. 1 in an iterative manner, where $Y^0$ are the initial labels, $t$ is the iteration count and $\alpha$ is a weighting hyperparameter. The propagation will converge upon infinite steps of propagation, i.e. $t \to \infty$ with a closed-form solution (Zhu et al., 2003).

$$Y^{t+1} = \alpha\tilde{W}Y^t + (1-\alpha)Y^0 \quad \Rightarrow \quad Y^\infty = (\mathbb{I} - \alpha\tilde{W})^{-1}Y^0 \qquad (1)$$

Under our VLM adaptation protocol, the initial label matrix is as follows, where $Y_p, Y_l, Y_u$ refer to the label of textual prototypes, few-shot labeled samples and unlabeled samples.

$$Y^0 = [Y_p^0, Y_l^0, Y_u^0], \ s.t. \ Y_p^0 = \mathbf{diag}(\mathbf{1}^{N_p}), \ Y_l^0 \in \{0,1\}^{N_l \times C}, \ Y_{lic}^0 = \mathbb{1}(y_i = c), \ Y_u^0 = \mathbf{0}^{N_u \times C}$$
$$(2)$$

**Efficient Label Propagation via Iterative Solution**: The closed-form solution to label propagation was adopted by Kalantidis et al. (2024). However, we argue that this closed-form solution is suboptimal in practice for two reasons. First, the closed-form solution requires solving a linear system using the conjugate gradient method (Grady, 2006). This is an expensive step that prohibits efficient inference. Furthermore, the closed-form solution is the converged solution to the iterative method. Each iteration step involves propagating labels between all connected nodes, including the inferred label information from test samples back to the text prototypes and few-shot labeled samples, which is undesirable. For this reason, we employ the iterative solution and we reset the labels for textual prototypes and labeled samples after each iteration, i.e. $Y_p^{t+1} = Y_p^0, Y_l^{t+1} = Y_l^0$.

## 3.3 DYNAMIC GRAPH EXPANSION FOR INDUCTIVE INFERENCE

In this section, we introduce the graph construction process. In particular, **Static Graph Construction** elaborates how a graph is constructed from all observed data samples and text prompts, and **Dynamic Graph Expansion** introduces how to efficiently expand the graph upon observing new testing data. Finally, we introduce a stream-based **Incremental Label Propagation** method for inductive inference.

**Static Graph Construction**: We first introduce the way to construct a static graph when the whole set of unlabeled test samples $\mathcal{D}_u$, textual prototypes $\mathcal{P}$, and few-shot samples (optional) $\mathcal{D}_l$ are available. Nodes are denoted as $\mathcal{V} = \{u_1, ..., u_{N_u}, p_1, ..., p_{N_p}, l_1, ..., l_{N_l}\}$. We write the adjacency matrix $W$ blockwise in Eq. 3 where $W_u \in \mathbb{R}^{N_u \times N_u}$, $W_p \in \mathbb{R}^{N_p \times N_p}$, $W_l \in \mathbb{R}^{N_l \times N_l}$, $W_{up} \in \mathbb{R}^{N_u \times N_p}$, $W_{ul} \in \mathbb{R}^{N_u \times N_l}$, $W_{pl} \in \mathbb{R}^{N_p \times N_l}$. We do not connect any nodes between textual prototypes and few-shot samples because we are only interested in inferring the labels of unlabeled test samples. This results in $W_p = \mathbf{0}, W_l = \mathbf{0}, W_{pl} = \mathbf{0}$.

$$W = \begin{bmatrix} W_u & W_{up} & W_{ul} \\ W_{up}^\top & W_p & W_{pl} \\ W_{ul}^\top & W_{pl}^\top & W_l \end{bmatrix} \Rightarrow W = \begin{bmatrix} W_u & W_{up} & W_{ul} \\ W_{up}^\top & \mathbf{0} & \mathbf{0} \\ W_{ul}^\top & \mathbf{0} & \mathbf{0} \end{bmatrix} \qquad (3)$$

For label propagation, we sparsify $W$ to improve computation cost and robustness. Observations made in Zhu et al. (2023); Kalantidis et al. (2024) suggest that cosine similarities between inputs from different modalities can vary significantly. For example, the cosine similarity between the features of two images is generally significantly lower than between an image feature and a text embedding. Therefore, we search for the K-nearest neighbors within each modality separately as follows, with $\text{NN}k_{\mathcal{D}}$ referring to the $k$ nearest neighbor search within nodes from set $\mathcal{D}$.

$$W_{ij} = \begin{cases} u_i^\top u_j, & \text{if } u_j \in \text{NN}k_{\mathcal{D}_u}(u_i) \\ u_i^\top p_j, & \text{if } p_j \in \text{NN}k_{\mathcal{P}}(u_i) \\ u_i^\top l_j, & \text{if } l_j \in \text{NN}k_{\mathcal{D}_l}(u_i) \text{ and } \mathcal{D}_l \neq \emptyset \\ 0 & \text{otherwise} \end{cases} \tag{4}$$

**Dynamic Graph Expansion**: The static graph construction assumes unlabeled test samples are all available at once. This assumption prohibits inductive inference where test samples arrive in a stream fashion. Previous work (Kalantidis et al., 2024) proposed an inductive inference approach by exploiting an external training dataset to build the manifold. With access to such an external dataset, a naive way to support inductive inference is to build a new graph from scratch upon seeing new testing data, which results in high computation complexity. For example, with a total number of $N_v = N_p + N_l + N_u$ nodes, each with $d$ dimensions, the computational complexity is $O(d \cdot N_v^3 + N_v^2 \log N_v)$, which prohibits realistic inference on large scale downstream task. To enable inductive inference while exploiting the testing data manifold, we reduce the complexity to $O(d \cdot N_v^2)$ by a dynamic graph expansion mechanism. Specifically, with an existing adjacency matrix $W$, we use the new test sample $u_{N_u+1}$ to query all existing nodes and replace the weakest connections with connections to the new test sample following the update rule in Eq. 5. The update rule will insert a new test sample into the existing graph without re-calculating the K nearest neighbor of the existing nodes.

$$\begin{aligned} \forall i,j \in 1 \cdots N_u, \quad & S_i = u_{N_{u+1}}^\top u_i, \\ W_{iN_{u+1}} = S_i \cdot \mathbb{1}(S_i > \min_j W_{ij}), \quad & W_{ij} = W_{ij} \cdot \mathbb{1}(S_i > \min_j W_{ij}) \cdot \mathbb{1}(j \neq \arg\min_j W_{ij}) \end{aligned} \tag{5}$$

**Incremental Label Propagation**: With the dynamically expanded graph, we can perform label propagation incrementally. Specifically, we re-use the pseudo labels from the previous iteration to speed up the convergence of the label propagation process. After inference for each test sample, the pseudo labels are attenuated for label propagation for the next test sample. We follow the rules in Eq. 6 to produce the incremental labels $Y_u^0$ for the observed test samples for inference on future test samples, where $Y_u^T$ represents the label predictions obtained after $T$ propagation steps.

$$\hat{Y}_{ui} = \begin{cases} \beta Y_{uic}^T, & \text{if } c = \arg\max_c Y_{uic}^T \\ 0, & \text{otherwise.} \end{cases}, \quad Y_u^0 = [\hat{Y}_u, \mathbf{0}^{1\times C}] \tag{6}$$

### 3.4 CONTEXT-AWARE EDGE RE-WEIGHTING

The graph construction process depends on a properly chosen distance metric. Measuring the distance between samples based on the features from the vision encoder is subject to the biases learnt by the vision encoder during large-scale pre-training. For example, it is known that VLMs encode features that capture all visual instances in the image, e.g. objects, background, style, etc. For this reason, directly measuring the cosine similarity between the raw features encoded by the VLM is subject to irrelevant information for the downstream task. For example, adapting the VLM to a downstream task of car model classification requires less attention to certain aspects like background or color information. Motivated by this, we propose to re-weight the importance of VLM vision-encoded features for calculating similarity. Specifically, we first calculate the statistics of textual prompts feature dimensions as follows, where $c$ is the feature dimension index.

$$\mu_c^p = \frac{1}{|\mathcal{P}|} \sum_i p_{ic}, \quad \sigma_c^p = \frac{1}{|\mathcal{P}|} \sum_i (p_{ic} - \mu_c)^2 \tag{7}$$

A higher variance in $c$-th dimension suggests the higher discriminative capability of the dimension, i.e. the more capable it is to discriminate between text prompts. Therefore, we increase the importance of these dimensions with higher variance. Additionally, we calculate the same statistics on the few-shot samples, resulting in $\mu_c^l$ and $\sigma_c^l$. In contrast to the context of textual prompts, we argue that high-variance feature dimensions in the context of few-shot samples highlight the intra-class variance and that this should be suppressed when calibrating the similarity. Therefore, we use the reciprocal of the variance for re-weighting graph edge similarity. The new edge weights are presented in Eq. 8:

$$W_{ij}^u = u_i^\top Norm(\mathbf{diag}(\sigma^p)u_j), \quad W_{ij}^{lu} = l_i^\top Norm(\mathbf{diag}(1/\sigma^l)u_j) \tag{8}$$

**Graph Sparsification**: The constructed graph via KNN search and similarity re-weighting via feature importance cater to the requirements of VLM adaptation. To further sparsify the graph to prune out erroneous connections and highlight the difference between different semantic classes in the downstream task, we further apply a power operation on the affinity matrix as $W_{ij} = W_{ij}^\gamma$.

## 3.5 OVERALL ALGORITHM

We present the overall algorithm for unified iterative label propagation for vision-language models in Algorithm 1.

---

**Algorithm 1** Iterative Label Propagation for Adapting Vision-Language Model

---

1: **Input:** Test data stream $\mathcal{D}_u$, textual prototypes $\mathcal{P}$, few-shot features $\mathcal{D}_l$, and label propagation iterations $T$
2: **Output:** Predicted labels $\{y_i\}$
3: Initial graph $W = \mathbf{0}$, initial label $Y_p^0, Y_l^0$ & $Y_u^0$ by Eq. 2 # Initialization
4: **for** $x_i \in \mathcal{D}_u$ **do**
5:   Calculate $W_{ij}^u$, and $W_{ij}^{ul}$ according to Eq. 8 # Testing Node Edge Update
6:   Update $W_u, W_{up}$ and $W_{ul}$ according to Eq. 5 # Dynamic Graph Expansion
7:   $W = W + W^\top, W = W^\gamma, \tilde{W} = D^{-\frac{1}{2}} W D^{-\frac{1}{2}}$ # Symmetrize, sparsify & normalize graph
8:   **for** $t = 0$ **to** $(T-1)$ **do**
9:    $Y^{t+1} = \tilde{W} Y^t$ # One step label propagation
10:    $Y_p^{t+1} = Y_p^0, Y_l^{t+1} = Y_l^0$ # Reset prototype and few-shot labels
11:   **end for**
12:   $y_i = \arg\max_c Y_{ic}^u$ # Test sample label
13:   Update $Y_u^0$ by eq. 6
14: **end for**
15: **return** $\{y_i\}$

---

## 4 EXPERIMENT

### 4.1 EXPERIMENTAL SETTINGS

**Datasets:** We selected a diverse range of datasets for thorough evaluation, grouped into three main categories: fine-grained datasets, style-transfer datasets, and out-of-distribution datasets, which cover a total of 30 specific test scenarios. **Fine-Grained Datasets:** We evaluated 11 widely recognized benchmarks, including ImageNet (Deng et al., 2009), Flowers102 (Nilsback & Zisserman, 2008), DTD (Cimpoi et al., 2014), OxfordPets (Parkhi et al., 2012), StanfordCars (Krause et al., 2013), UCF101 (Soomro, 2012), Caltech101 (Fei-Fei et al., 2004), Food101 (Fei-Fei et al., 2004), SUN397 (Xiao et al., 2010), FGVCAircraft (Maji et al., 2013), and EuroSAT (Helber et al., 2019). **Style-Transfer Datasets:** We used five benchmarks: ImageNet-V2 (Recht et al., 2019), ImageNet-A (Hendrycks et al., 2021b), ImageNet-R (Hendrycks et al., 2021a), and ImageNet-Sketch (Wang et al., 2019). **Out-of-Distribution Datasets:** We extensively assessed 15 unique different domains

from ImageNet-C (Hendrycks & Dietterich, 2019) with the highest severity 5, including Gaussian Noise, Shot Noise, Impulse Noise, Defocus Blur, Frosted Glass Blur, Motion Blur, Zoom Blur, Snow, Frost, Fog, Brightness, Contrast, Elastic Transform, Pixelate, and JPEG corruptions.

**Implementation Details:** For all experiments, we use the pretrained CLIP model (Radford et al., 2021) with both ResNet-based and ViT-based architectures, specifically ResNet50 (He et al., 2016) and ViT-B/16 (Dosovitskiy, 2020). Unlike previous works, which tuned hyperparameters separately for each experiment, we simplify the process by using a consistent set of hyperparameters across all experiments. In constructing the KNN graph, each test image is connected to 3 text embeddings, 8 embeddings from the test memory bank, and 8 from few-shot images (if applicable). The exponential factor $\gamma$ for graph sparsification is set to 10, the update factor $\beta$ for $\hat{Y}_{ui}$ is set to 0.2, and the factor $\alpha$ for label propagation is set to 1.0. For efficient inference, we limit the label propagation iterations $T$ to 3. We also adopt the dataset splits and textual prompts from Pratt et al. (2023); Zhang et al. (2022).

**Competing Methods:** We selected a diverse set of competing methods for both zero-shot and few-shot adaptation scenarios to create a comprehensive benchmark. These methods can be broadly categorized into two groups: prompt-tuning methods and adapter-based methods. For prompt tuning, we include TPT (Shu et al., 2022), DiffTPT (Feng et al., 2023), CoOp (Zhou et al., 2022b), CoCoOp (Zhou et al., 2022a), among others. For adapter-based methods, we consider CLIP-Adapter (Gao et al., 2024), TIP-Adapter (Zhang et al., 2022), and the most recent approaches such as DMN (Zhang et al., 2024b), TDA (Karmanov et al., 2024), and ZLaP (Kalantidis et al., 2024), along with several other methods. Finally, we benchmark our proposed method, ECALP, on all tasks.

## 4.2 Evaluation of Zero-shot Adaptation

**Fine-Grained Categorization Tasks**: We present the results of zero-shot adaptation of CLIP model to fine-grained categorization downstream tasks in Tab. 1. We make the following observations from the results. **i)** Our proposed method consistently outperforms other methods across both architectures, showcasing robust generalization and strong performance, especially in the ViT-B/16 architecture. The ViT-based models generally perform better than their ResNet-based counterparts, reflecting the advantages of transformer-based architectures in handling diverse benchmarks. **ii)** DMN performs strongly across both architectures, particularly excelling in fine-grained classification tasks like Pets and Aircraft. However, the original DMN, denoted as DMN* in the table, implements exhaustive searching for the fusion hyperparameters based on testing set for the individual downstream tasks and the performance drops by more than 1% when hyperparameter searching is removed. Nevertheless, both DMN* and its variant, DMN with fixed hyperparameters, are worse in general compared with us. Also, ZLaP$^\dagger$ requires additional unlabelled training set data to initialize the graph. **iii)** Some methods show strengths in specific benchmarks. For example, DiffPT excels in handling corruption datasets. The state-of-the-art label propagation based method, ZLap, achieves much lower performance compared with us. This is attributed to the more effective context-aware reweighting and iterative propagation strategies.

**Style-Transfer Tasks**: We further evaluate the performance on style-transfer downstream tasks with results presented in Tab. 2. Our analysis reveals several key insights. **i)** Our proposed method consistently surpasses the baseline CLIP models, such as CLIP-RN50 and CLIP-ViT/B-16, with significant improvements across all datasets, especially in the averaged scores, with an average gain of approximately 7%. This highlights the effectiveness of our approach in improving generalization without the need for additional training. **ii)** In comparison to state-of-the-art adaptation methods such as DMN and TDA, our method achieves higher accuracy averaged over all datasets, demonstrating that our method is robust across different architectures. iii) Remarkably, even when compared to adaptation methods that require training, such as CoOp and CoCoOp, our method maintains a clear advantage, showcasing the superior performance of our method, even in scenarios where competing methods undergo task-specific fine-tuning.

**Out-of-Distribution Tasks**: Finally, we evaluate the model's adaptation to out-of-distribution downstream tasks. As seen from Tab. 3, our proposed method consistently outperforms state-of-the-art competing methods on almost all types of corruptions with both CNN and transformer backbones.

Table 1: Zero-shot adaptation of CLIP on fine-grained categorization downstream tasks. DMN* refers to the original method that exhaustively searches the fusion hyperparameter with testing data ground-truth on each individual downstream task (not for direct comparison with other methods). ZLaP[†] requires additional unlabelled training set data.

| Method | ImageNet | Flower | DTD | Pets | Cars | UCF | Caltech | Food | SUN | Aircraft | EuroSAT | Mean |
|---|---|---|---|---|---|---|---|---|---|---|---|---|
| CLIP-RN50 (Radford et al., 2021) | 58.16 | 61.75 | 40.37 | 83.57 | 55.70 | 58.84 | 85.88 | 73.97 | 58.80 | 15.66 | 23.69 | 56.04 |
| DN (Zhou et al., 2023) | 60.16 | 63.32 | 41.21 | 81.92 | 56.55 | 55.60 | 87.25 | 74.64 | 59.11 | 17.43 | 28.31 | 56.86 |
| TPT (Shu et al., 2022) | 60.74 | 62.69 | 40.84 | 84.49 | 58.46 | 60.82 | 87.02 | 74.88 | 61.46 | 17.58 | 28.33 | 57.94 |
| DiffTPT (Feng et al., 2023) | 60.80 | 63.53 | 40.72 | 83.40 | **60.71** | 62.67 | 86.89 | **79.21** | 62.72 | 17.60 | 41.04 | 59.94 |
| VisDesc (Menon & Vondrick, 2023) | 59.68 | 65.37 | 41.96 | 82.39 | 54.76 | 58.47 | 88.11 | 76.80 | 59.84 | 16.26 | 37.60 | 58.29 |
| Ensemble (Zhang et al., 2022) | 60.32 | 66.10 | 40.07 | 85.83 | 55.71 | 61.33 | 83.94 | 77.32 | 58.53 | 17.10 | 37.54 | 58.53 |
| CALIP (Guo et al., 2023) | 60.57 | 66.38 | 42.39 | 86.21 | 56.27 | 61.72 | 87.71 | 77.42 | 58.59 | 17.76 | 38.90 | 59.45 |
| CuPL (Pratt et al., 2023) | 61.45 | 65.44 | 48.64 | 84.84 | 57.28 | 58.97 | 89.29 | 76.94 | 62.55 | 19.59 | 38.38 | 60.31 |
| SuS-X (Udandarao et al., 2023) | 61.84 | 67.72 | 50.59 | 85.34 | 57.27 | 61.54 | 89.53 | 77.58 | 62.95 | 19.47 | 45.57 | 61.76 |
| DMN (Zhang et al., 2024b) | 62.02 | 68.33 | 50.53 | 86.29 | 58.36 | 64.02 | 89.09 | 74.69 | 63.70 | 20.22 | 44.94 | 62.02 |
| DMN* (Zhang et al., 2024b) | 63.87 | 67.93 | 50.41 | 86.78 | 60.02 | 65.34 | 90.14 | 76.70 | 64.39 | 22.77 | 48.72 | 63.37 |
| TDA (Karmanov et al., 2024) | 61.35 | 68.74 | 43.74 | 86.18 | 57.78 | 64.18 | 89.70 | 77.75 | 62.53 | 17.61 | 42.11 | 61.06 |
| ZLaP[†] (Kalantidis et al., 2024) | 62.20 | 69.27 | 42.79 | 80.32 | 56.42 | 62.81 | 86.90 | 77.87 | 61.83 | 17.37 | 31.85 | 59.06 |
| ECALP (Ours) | **62.64** | **69.39** | **54.49** | **88.20** | 60.56 | **66.67** | **89.94** | 76.97 | **64.97** | 21.12 | 49.09 | **64.00** |
| CLIP-ViTB/16 (Radford et al., 2021) | 66.73 | 64.44 | 44.27 | 88.25 | 65.48 | 65.13 | 93.35 | 83.65 | 62.59 | 23.67 | 42.01 | 63.87 |
| Ensemble (Zhang et al., 2022) | 68.34 | 66.99 | 45.04 | 86.92 | 66.11 | 65.16 | 93.55 | 82.86 | 65.63 | 23.22 | 50.42 | 64.93 |
| TPT (Shu et al., 2022) | 68.98 | 68.98 | 47.75 | 87.79 | 66.87 | 68.04 | 94.16 | 84.67 | 65.50 | 24.78 | 42.44 | 65.45 |
| DiffTPT (Feng et al., 2023) | 70.30 | 70.10 | 47.00 | 88.20 | 67.01 | 68.22 | 92.49 | **87.23** | 65.74 | 25.60 | 43.13 | 65.90 |
| DMN (Zhang et al., 2024b) | 70.51 | 75.32 | 54.85 | 91.22 | 67.01 | 71.95 | 93.63 | 84.05 | 69.14 | 28.29 | 56.22 | 69.29 |
| DMN* (Zhang et al., 2024b) | 72.25 | 74.49 | 55.85 | 92.04 | 67.96 | 72.51 | 95.38 | 85.08 | 70.18 | 30.03 | 59.43 | 70.47 |
| TDA (Karmanov et al., 2024) | 69.51 | 71.42 | 47.40 | 88.63 | 67.28 | 70.66 | 94.24 | 86.14 | 67.62 | 23.91 | **58.00** | 67.71 |
| ZLaP[†] (Kalantidis et al., 2024) | 70.17 | 73.49 | 48.58 | 87.14 | 65.63 | 71.45 | 93.06 | 86.92 | 67.44 | 25.44 | 55.62 | 67.72 |
| ECALP (Ours) | **71.26** | **75.96** | **56.32** | **92.31** | **68.20** | **75.44** | **94.40** | 85.72 | **70.35** | **29.49** | 56.53 | **70.54** |

Table 2: Zero-shot adaptation of CLIP on style-transfer downstream tasks. Methods marked with [†] are pre-finetuned using additional 16-shot training samples for each category in ImageNet.

| Method | ImageNet-A | ImageNet-V2 | ImageNet-R | ImageNet-S | Mean |
|---|---|---|---|---|---|
| CLIP-RN50 (Radford et al., 2021) | 21.83 | 51.41 | 56.15 | 33.37 | 40.69 |
| CoOp[†] (Zhou et al., 2022b) | 23.06 | 55.40 | 56.60 | 34.67 | 42.43 |
| CoCoOp[†] (Zhou et al., 2022a) | 23.32 | 55.72 | 57.74 | 34.48 | 42.82 |
| TPT (Shu et al., 2022) | 26.67 | 54.70 | 59.11 | 35.09 | 43.89 |
| DiffTPT (Feng et al., 2023) | **31.06** | 55.80 | 58.80 | 37.10 | 45.69 |
| CALIP (Guo et al., 2023) | 23.96 | 53.70 | 60.81 | 35.61 | 43.52 |
| TDA (Karmanov et al., 2024) | 30.29 | 55.54 | 62.58 | 38.12 | 46.63 |
| DMN (Zhang et al., 2024b) | 28.57 | 56.12 | 61.44 | 39.84 | 46.49 |
| ECALP (Ours) | 28.80 | **56.92** | **63.68** | **41.51** | **47.73** |
| CLIP-ViT-B/16 (Radford et al., 2021) | 47.87 | 60.86 | 73.98 | 46.09 | 57.20 |
| CoOp[†] (Zhou et al., 2022b) | 49.71 | 64.20 | 75.21 | 47.99 | 59.14 |
| CoCoOp[†] (Zhou et al., 2022a) | 50.63 | 64.07 | 76.18 | 48.75 | 59.91 |
| MaPLe[†] (Khattak et al., 2023) | 50.90 | 64.07 | 76.98 | 49.15 | 60.28 |
| TPT (Shu et al., 2022) | 54.77 | 63.45 | 77.06 | 47.94 | 60.81 |
| DiffTPT (Feng et al., 2023) | 55.68 | 65.10 | 75.00 | 46.80 | 60.65 |
| TDA (Karmanov et al., 2024) | **60.11** | 64.67 | 80.24 | 50.54 | 63.89 |
| DMN (Zhang et al., 2024b) | 58.28 | 65.17 | 78.55 | 53.20 | 63.80 |
| ECALP (Ours) | 58.52 | **65.72** | **80.77** | **54.66** | **64.92** |

Table 3: Zero-shot adaptation of CLIP on out-of-distribution downstream tasks. ZLaP[†] requires additional unlabelled training set data.

| Method | Gauss. | Shot | Impu. | Defo. | Glas. | Moti. | Zoom | Snow | Fros. | Fog | Brig. | Cont. | Elas. | Pix. | JPEG | Average |
|---|---|---|---|---|---|---|---|---|---|---|---|---|---|---|---|---|
| CLIP-RN50 (Radford et al., 2021) | 1.63 | 2.18 | 1.64 | 10.06 | 3.42 | 7.85 | 12.83 | 12.58 | 15.67 | 21.95 | 40.27 | 6.28 | 4.75 | 11.12 | 13.03 | 11.02 |
| TDA (Karmanov et al., 2024) | 2.26 | 3.10 | 2.31 | 11.30 | 5.12 | 9.26 | 15.43 | 15.47 | 19.11 | 26.45 | 45.30 | 8.34 | 7.30 | 13.01 | 15.83 | 13.31 |
| ZLaP[†] (Kalantidis et al., 2024) | 1.77 | 2.33 | 1.65 | 10.30 | 3.54 | 7.99 | 13.47 | 13.66 | 17.15 | 23.20 | 44.67 | 6.55 | 5.15 | 11.61 | 14.23 | 11.82 |
| DMN (Zhang et al., 2024b) | 2.14 | 2.78 | 2.30 | 10.91 | 4.48 | 8.59 | 14.31 | 14.14 | 17.92 | 24.16 | 44.57 | 7.88 | 6.11 | 12.40 | 14.93 | 12.51 |
| ECALP (Ours) | **2.71** | **3.30** | **2.82** | **12.29** | **5.49** | **10.56** | **16.82** | **16.66** | **20.60** | **27.83** | **47.02** | **9.08** | **7.72** | **14.46** | **16.88** | **14.28** |
| CLIP-ViTB/16 (Radford et al., 2021) | 11.34 | 12.31 | 11.85 | 23.78 | 15.12 | 24.05 | 22.72 | 32.70 | 30.43 | 36.69 | 54.57 | 16.84 | 12.77 | 31.22 | 33.00 | 24.63 |
| TDA (Karmanov et al., 2024) | 15.42 | 16.46 | 16.03 | 26.53 | 17.91 | 27.35 | 25.90 | 36.50 | 34.84 | 40.53 | 58.57 | 20.16 | 16.62 | 35.65 | 36.69 | 28.34 |
| ZLaP[†] (Kalantidis et al., 2024) | 12.83 | 14.03 | 13.27 | 24.88 | 16.13 | 25.77 | 24.36 | 34.43 | 32.63 | 38.56 | 58.42 | 17.53 | 14.21 | 33.72 | 35.52 | 26.42 |
| DMN (Zhang et al., 2024b) | 14.33 | 15.33 | 14.69 | 26.06 | 17.19 | 26.61 | 25.23 | 34.81 | 33.48 | 38.93 | 58.70 | 19.38 | 15.40 | 35.32 | 36.49 | 27.46 |
| ECALP (Ours) | **15.92** | **16.84** | **16.32** | **27.85** | **18.78** | **28.59** | **27.62** | **37.82** | **36.01** | **41.65** | **60.57** | **21.26** | **17.77** | **37.39** | **38.11** | **29.50** |

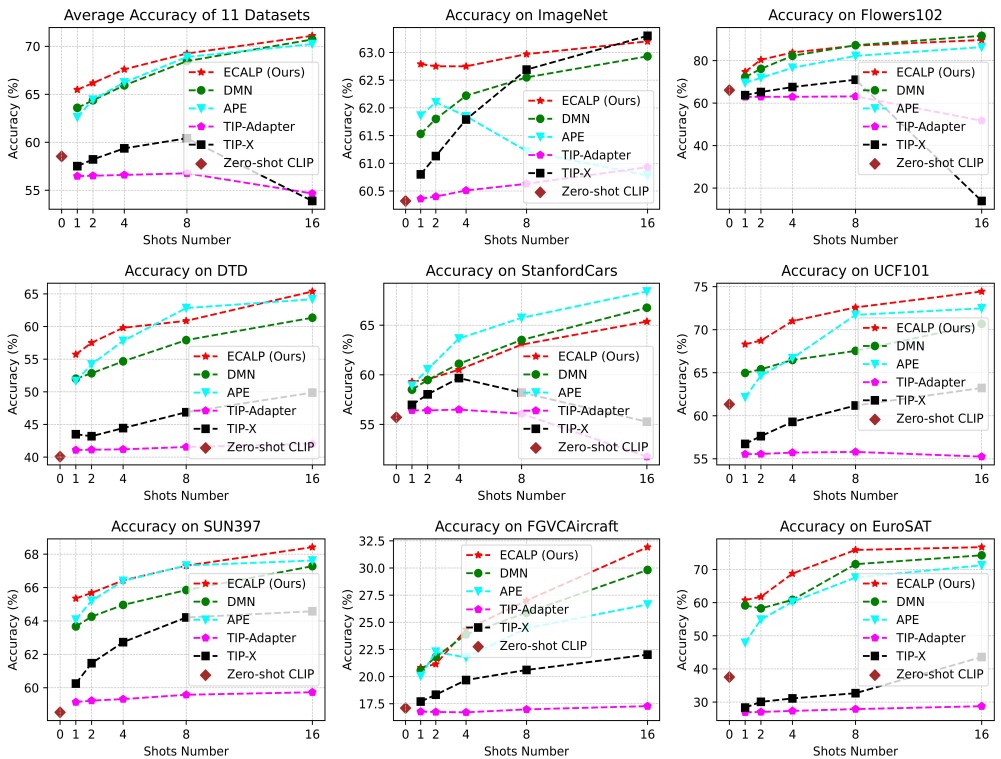

Figure 2: Few-shot adaptation of VLM for fine-grained categorization downstream tasks with CLIP-RN50 model.

## 4.3 EVALUATION OF FEW-SHOT ADAPTATION

For few-shot adaptation tasks, we evaluate the adaptation performance on fine-grained categorization benchmarks, with [1, 2, 4, 8, 16] shots of each class, as presented in Fig. 2. In line with (Silva-Rodriguez et al., 2024), we establish a fair benchmark by comparing existing methods using their default hyperparameter values, as specified in the corresponding papers, without conducting a hyperparameter search. We compare our method with several existing training-free approaches, including DMN(Zhang et al., 2024b), APE (Zhu et al., 2023), TIP-Adapter (Zhang et al., 2022), and TIP-X (Udandarao et al., 2023). Our method outperforms all competing methods on average across 11 datasets and consistently achieves superior results across most datasets and shot counts. Furthermore, it maintains high performance even in low-shot scenarios, underscoring its efficiency and robustness.

## 4.4 ABLATION & ADDITIONAL STUDY

**Unveiling the Impact of Individual Components**: We first conduct a comprehensive ablation analysis investigating the effectiveness of individual components within ECALP. As shown in Tab. 4, we first observe that label propagation (Label Prop.) improves the results with a significant margin, on DTD and UCF datasets, compared to naive nearest neighbor classification. However, the result drop a little bit on ImageNet (58.16% → 55.34%), probably owing to the diversity of contents within ImageNet. This is quickly remedied when edge weights are reweighted by text embeddings (Tex. Reweight) (55.34% → 62.64%), suggesting the context information is essential to identify the relevant information from the feature encoded by VLM. When 16 shots of each class are available, we make similar observations that edge weights re-weighted by few-shot sample feature distribution (F.S. Reweight) are essential to maintain good performance while text reweighting is less important. We also noticed that ImageNet benefits less from few-shot than DTD and UCF because of the high intra-class variation, which renders few-shot labeled samples less effective.

Table 4: Ablation study on the components of ECALP. (ZS: zero-shot; FS: few-shot)

| Strategy | Lab. Prop. | Few-shot | Tex. Reweight | F.S Reweight | ImageNet | DTD | UCF |
|---|---|---|---|---|---|---|---|
| - | ✓ | - | - | - | 55.34 | 49.76 | 61.17 |
| ECALP-ZS | ✓ | - | ✓ | - | 62.64 | 54.49 | 66.67 |
| - | ✓ | ✓ | - | - | 47.25 | 59.34 | 66.98 |
| - | ✓ | ✓ | - | ✓ | 63.47 | 65.13 | 74.18 |
| ECALP-FS | ✓ | ✓ | ✓ | ✓ | 63.20 | 65.37 | 74.44 |

Table 5: Ablation study on label propagation.

| Strategy | $W_{up}$ | $W_u$ | $W_{lu}$ | ImageNet | DTD | UCF |
|---|---|---|---|---|---|---|
| - | ✓ | - | - | 62.12 | 52.42 | 63.97 |
| ECALP-ZS | ✓ | ✓ | - | 62.64 | 54.49 | 66.67 |
| - | ✓ | - | ✓ | 63.00 | 64.48 | 72.01 |
| ECALP-FS | ✓ | ✓ | ✓ | 63.20 | 65.37 | 74.44 |

**Investigating Graph Construction**: We further investigate the necessity of each subgraph of the overall graph. Specifically, we consider the three subgraphs including test samples to prototypes $W_{up}$, within test samples $W_u$ and test samples to few-shot samples $W_{lu}$. As seen from Tab. 5, all three datasets benefit substantially from having the subgraph connecting to textual prototypes. When the subgraph within test samples $W_u$ is included, the performance further improves, though more significantly on DTD and UCF, suggesting the manifold is helpful to the propagation of labels. Finally, including the test sample subgraph $W_u$ is also helpful to the few-shot cases.

**Computational Efficiency.** To assess the computational efficiency of our proposed ECALP method, we measure the wall-clock time during the zero-shot adaptation task on fine-grained datasets, as shown in Tab. 6. All experiments are conducted using a single RTX 3090 GPU and an AMD EPYC 7302 CPU. ECALP, along with other training-free adaptation methods, does not significantly increase computational overhead and demonstrates at least a $30\times$ speedup compared to approaches requiring training. Notably, ECALP achieves the best performance with low computational cost.

Table 6: Comparison of wall-clock time with CLIP-RN50.

| Method | ImageNet | | | DTD | | | UCF | | |
|---|---|---|---|---|---|---|---|---|---|
| | Testing Time | Accuracy | Gain | Testing Time | Accuracy | Gain | Testing Time | Accuracy | Gain |
| CLIP-RN50 (Radford et al., 2021) | 14.0ms | 58.16 | 0.00 | 10.1ms | 40.37 | 0.00 | 9.4ms | 58.84 | 0.00 |
| Training-required | | | | | | | | | |
| TPT (Shu et al., 2022) | 898.7ms | 60.74 | +2.58 | 881.9ms | 40.84 | +0.47 | 871.6ms | 60.82 | +1.98 |
| DiffTPT (Feng et al., 2023) | 2472.5ms | 60.80 | +2.64 | 2359.4ms | 40.72 | +0.35 | 2115.7ms | 62.67 | +3.83 |
| Training-free | | | | | | | | | |
| TDA (Karmanov et al., 2024) | 26.7ms | 61.35 | +3.19 | 22.0ms | 43.74 | +3.37 | 13.3ms | 64.18 | +5.34 |
| DMN (Zhang et al., 2024b) | 30.1ms | 62.02 | +3.86 | 18.8ms | 50.53 | +10.16 | 15.5ms | 64.02 | +5.18 |
| ZLaP (Kalantidis et al., 2024) | 29.3ms | 62.20 | +4.04 | 15.1ms | 42.79 | +2.42 | 13.4ms | 62.81 | +3.34 |
| ECALP (Ours) | 28.0ms | 62.64 | +4.47 | 16.1ms | 54.49 | +14.12 | 14.5ms | 66.67 | +7.83 |

## 5 CONCLUSION

Adapting pre-trained vision language model for downstream tasks introduces a new paradigm for crafting computer vision models. We aim to better exploit the observed unlabeled data, i.e. the data manifold and propose a unified method based on label propagation. In particular, we addressed the challenges of improving the computation efficiency of label propagation via iterative and incremental solution. We also proposed a simple dynamic graph expansion strategy to accommodate inductive inference. Furthermore, we observe that the similarity metric adopted by existing methods overlooked the diverse information captured by VLM and thus propose a context-aware edge reweighting based on the downstream task information. We carried out experiments on fine-grained categorization and out-distribution downstream tasks and achieved the state-of-the-art performance on all datasets.

## ACKNOWLEDGEMENTS

This work was supported by the National Natural Science Foundation of China (NSFC) under Grant 62106078; by the Agency for Science, Technology and Research, Singapore (A*STAR) under Grant M23L7b0021; by the Guangdong Provincial Key Laboratory of Human Digital Twin under Grant 2022B1212010004; by the National Natural Science Foundation of China (NSFC) under Grant 61701181; by the Guangdong R&D key project of China under Grant 2019B010155001.

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

# A APPENDIX

## A.1 FURTHER ANALYSIS OF THE COMPUTATIONAL COMPLEXITY IN GRAPH CONSTRUCTION

In this section, we analyze the computational complexity of constructing a graph in two scenarios: first, the static graph construction, and then, how our dynamic graph expansion method significantly reduces this complexity. The analysis focuses on two key components: **distance matrix computation** and **nearest neighbor selection**. For simplicity, we assume the graph consists of a total of $N_v = N_p + N_l + N_u$ nodes, where each node is represented by a feature vector of dimension $d$. Since this analysis is conducted in the context of inductive inference, we consider the process as a sequential addition of nodes from 1 to $N_v$.

### A.1.1 STATIC GRAPH CONSTRUCTION COMPLEXITY

**Distance Computation Complexity:** In a K-Nearest Neighbors (KNN) graph, the first step is to compute the pairwise distances between nodes. For $n$ nodes, each represented by a $d$-dimensional feature vector, the time complexity to compute the distance between two nodes is $O(d)$. Since distances need to be calculated for all pairs of nodes, the number of comparisons is $O(n^2)$. Hence, the computational complexity for distance computation at step $n$ is:

$$C_{D_n} = O(n^2 \cdot d) \tag{9}$$

Accumulating the complexity across all steps, the total computational complexity for distance computation is:

$$C_D = \sum_{n=1}^{N_v} C_{D_n} = \sum_{n=1}^{N_v} O(n^2 \cdot d) = O\left(d \cdot \sum_{n=1}^{N_v} n^2\right) = O\left(d \cdot N_v^3\right) \tag{10}$$

**Neighbor Selection Complexity:** After computing the distances, the next step is to select the $K$-nearest neighbors for each node. Sorting the list of distances for a single node requires $O(n \log n)$ time. The total complexity for neighbor selection can be approximated by treating this as a continuous function and integrating:

$$C_S = O\left(\sum_{n=1}^{N_v} n \log n\right) \approx O\left(\int_1^{N_v} x \log x \, dx\right) = O\left(N_v^2 \log N_v\right) \tag{11}$$

**Total Complexity:** The overall time complexity for static graph construction is the sum of the complexities of distance computation and neighbor selection. Thus, the total time complexity is:

$$C = C_D + C_S = O(N_v^3 + N_v^2 \log N_v) \tag{12}$$

### A.1.2 DYNAMIC GRAPH EXPANSION COMPLEXITY

**Distance Computation Complexity:** In our dynamic graph expansion method, the distance computation for each new node is limited to the current node and the existing nodes in the graph. For $n$ nodes, the number of distance computations needed is $O(n \cdot d)$. Accumulating this across all steps, the total complexity for distance computation becomes:

$$C_D = \sum_{n=1}^{N_v} C_{D_n} = \sum_{n=1}^{N_v} O(n \cdot d) = O\left(d \cdot \sum_{n=1}^{N_v} n\right) = O\left(d \cdot N_v^2\right) \tag{13}$$

**Neighbor Selection Complexity:** In this dynamic method, we maintain a sparse affinity matrix, and only need to find the nearest neighbor for each node, which requires a constant time operation. Thus, the total complexity for neighbor selection is:

$$C_S = \sum_{n=1}^{N_v} O(1) = O(N_v) \tag{14}$$

**Total Complexity:** The overall time complexity for dynamic graph expansion is the sum of the distance computation and neighbor selection complexities. Therefore, the total time complexity is:

$$C = C_D + C_S = O(d \cdot N_v^2 + N_v) = O(d \cdot N_v^2) \tag{15}$$

## A.2 FURTHER ANALYSIS ON ECALP

### A.2.1 KNN CONNECTION STUDY

We conducted an analysis of the number of connections in K-Nearest Neighbors (KNN) models using CLIP-RN50 on the DTD dataset in Tab. 7. Specifically, we experimented with varying the number of connections for both NN$k_P$ and NN$K_{D_u}$, selecting values from the range [1, 3, 5, 8, 10]. It presents the performance across these different configurations, highlighting the highest accuracy achieved when $k_P = 3$ and $K_{D_u} = 8$, where the model reached an accuracy of 54.49%. This suggests that increasing the number of neighbors enhances performance up to a certain point, beyond which the impact diminishes.

Table 7: Studies on KNN connections number with CLIP-RN50 for zero-shot adaptation.

| $k_P \backslash K_{D_u}$ | 1 | 3 | 5 | 8 | 10 |
|---|---|---|---|---|---|
| 1 | 48.64 | 48.88 | 49.05 | 49.00 | 49.00 |
| 3 | 52.36 | 53.90 | 54.20 | **54.49** | 54.20 |
| 5 | 51.60 | 53.19 | 53.90 | 54.26 | 54.14 |
| 8 | 51.18 | 52.96 | 53.72 | 53.01 | 52.66 |
| 10 | 51.00 | 52.12 | 52.13 | 52.25 | 52.30 |

### A.2.2 DYNAMIC GRAPH CONSTRUCTION VS. STATIC

As illustrated in Fig. 3, our dynamic graph expansion enables ECALP to achieve an approximate $6\times$ speedup compared to the static graph construction method on the ImageNet dataset with CLIP-RN50.

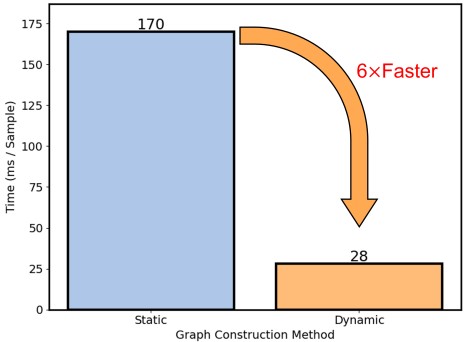

Figure 3: Wall-clock time comparison between statically-constructed graphs and dynamically-expansive graphs.

### A.2.3 GENERALIZATION TO DIFFERENT VLMS

The ability of ECALP to improve zero-shot adaptation extends beyond CLIP, showcasing its effectiveness across various Vision-Language Models (VLMs). As demonstrated in Tab. 8, applying

ECALP to multiple VLMs results in a noticeable improvement in ImageNet accuracy. For instance, when applied to BLIP (Li et al., 2022), the base ImageNet accuracy of 53.40% is significantly enhanced to 58.16%. Similarly, in the case of BLIP v2 (Li et al., 2023), the accuracy increases from 41.22% to 43.72% with ECALP. A notable improvement is also observed with ALBEF (Li et al., 2021), where ECALP raises the accuracy from 36.15% to 40.31%. These results highlight the generalization ability of our approach, making it applicable across different state-of-the-art VLMs, consistently improving their performance.

Table 8: Zero-shot adaptation accuracy with our method applied to different VLM.

| VLM | ImageNet Accuracy |
|---|---|
| BLIP (Li et al., 2022) | 53.40 |
| + ECALP (Ours) | **58.16** |
| BLIP v2 (Li et al., 2023) | 41.22 |
| +ECALP (Ours) | **43.72** |
| ALBEF (Li et al., 2021) | 36.15 |
| +ECALP (Ours) | **40.31** |

### A.2.4 COMBINATION WITH COOP

We integrate ECALP with the traditional prompt learning method CoOp (Zhou et al., 2022b), utilizing text embeddings with their trained prompts on 16-shot samples from the DTD dataset. As illustrated in Fig. 4, CoOp combined with ECALP significantly outperforms the original CoOp across different shot numbers. Notably, with 8 and 16 shots, CoOp with ECALP surpasses standalone ECALP, indicating that ECALP can be seamlessly integrated with traditional prompt learning methods to enhance their performance. However, in low-shot scenarios like 1-shot, CoOp with ECALP performs worse than ECALP alone. This is likely due to overfitting and bias in CoOp's trained prompts when training data is very limited, which is detrimental for label propagation using the complete test data manifold.

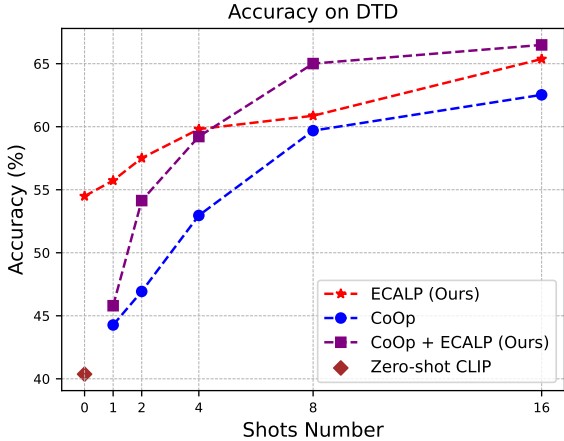

Figure 4: Combining our proposed ECALP with CoOp (Zhou et al., 2022b) on the DTD dataset.

### A.2.5 IMPACT OF LABEL PROPAGATION ITERATION $T$

We explored the effect of varying the label propagation iteration $T$ from 1 to 6 on the DTD dataset, as illustrated in Fig. 5. The results demonstrate that performing multiple iterations of label propagation enhances accuracy compared to a single iteration. This suggests that ECALP effectively utilizes the manifold structure of the test data. However, increasing iterations also raises computational costs, as $Y^{t+1}$ becomes denser with larger $t$, which may affect efficiency. Thus, we opt for $T = 3$ across

all experiments to maintain a balance between computational efficiency and accuracy. As indicated in Tab. 6, ECALP achieves an optimal trade-off between these factors.

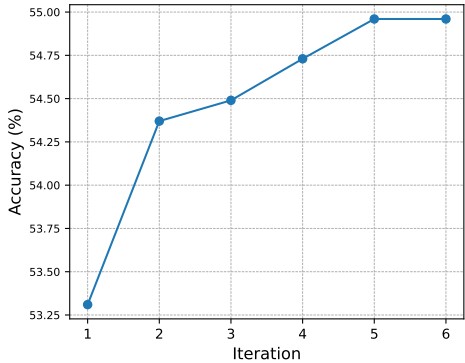

Figure 5: Effect of label propagation iterations on the DTD dataset.

### A.2.6 PERFORMANCE ON DIFFERENT CORRUPTION SEVERITIES

We further examine the performance of our proposed ECALP on corrupted downstream tasks using the ImageNet-C dataset, across severities ranging from 1 to 5. The average accuracy of ECALP compared to CLIP-ResNet50 over 15 types of corruption (as detailed in Tab. 3) is presented in Fig. 6. ECALP consistently surpasses CLIP by a margin of approximately 3%-5% across all severity levels, demonstrating its robustness.

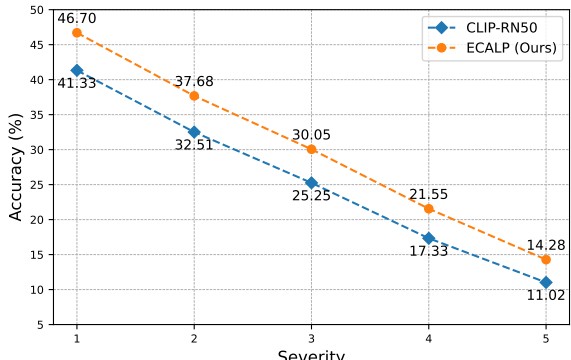

Figure 6: Average accuracy across 15 different corruption subsets under various severity levels on the ImageNet-C dataset.

### A.2.7 ROBUSTNESS TO POOR INITIALIZATION

To assess the robustness of our dynamically constructed graph for label propagation, we conducted experiments under various data-stream conditions. As presented in Tab. 9, we evaluated ECALP's performance with three different test set configurations: standard **Random Sampling**: A typical random sample of the test stream. **Hard Samples First**: The initial 5% of the test stream consists of only hard samples (those misclassified by CLIP), followed by a random sampling of the remaining 95%. **10% Hard Samples First**: The initial 10% of the test stream consists of only hard samples, with the rest being randomly sampled.

The results demonstrate a minimal performance drop, with less than a 0.4% decrease in accuracy even with a 10% hard sample initialization. This resilience can be attributed to our graph's dynamic nature, where edge connections between data points are continuously updated as the test sequence progresses. This adaptability ensures that our method remains robust against poor initializations and maintains high accuracy despite challenging conditions.

Table 9: Zero-shot adaptation accuracy with our method under different data-stream initializations on the DTD dataset.

|  | Random Sampling | 5% Hard Samples First | 10% Hard Samples First |
|---|---|---|---|
| Accuracy | 54.49 | 54.31 | 54.14 |

### A.2.8 VISUALIZATION SAMPLES FOR CORRUPTION TASK

Fig. 7 illustrates an image from the ImageNet-C (Hendrycks & Dietterich, 2019) dataset subjected to various types of corruption at severity level 5. The examples include Gaussian Noise, Motion Blur, Saturation, and Snow. These visualizations highlight the substantial domain shifts present in this challenging task, demonstrating the robustness required to handle such severe corruptions effectively.

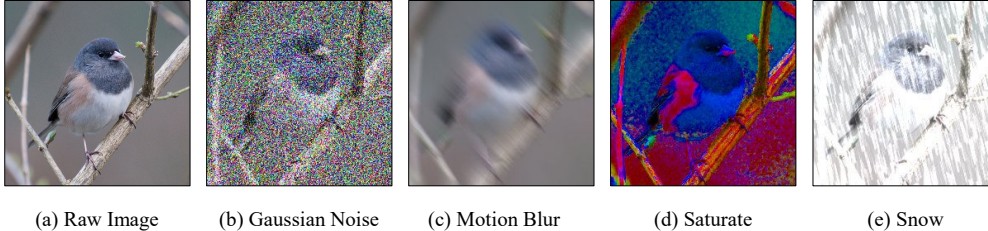

(a) Raw Image     (b) Gaussian Noise     (c) Motion Blur     (d) Saturate     (e) Snow

Figure 7: Samples from different corruption types at severity level 5 in the ImageNet-C dataset.

## A.3  FULL RESULTS FOR FEW-SHOT ADAPTATION

Figure 8: Full results for few-shot adaptation of VLM for fine-grained categorization downstream tasks.

## A.4  FAILURE CASE ANALYSIS FOR FEW-SHOT ADAPTATION

In this section, we delve into the reasons behind the suboptimal performance of our ECALP method on the OxfordPets and Food101 datasets under few-shot adaptation. We bring in other two datasets, Flowers102 and EuroSAT, for comparative studies. The test sample features are projected into 2D space via t-SNE (van der Maaten & Hinton, 2008), as illustrated in Fig. 9. We make the following observations. The test samples are more cluttered and mixed up in OxfordPets and Food101 than in Flowers102 and EuroSAT. There are more separated sub-classes, i.e. isolated clusters within each semantic class, for OxfordPets and Food101. This suggests having a few-shot labeled samples is less effective for OxfordPets and Food101 datasets. In certain cases, an inappropriate selection of few-shot samples may even bias the adaptation.

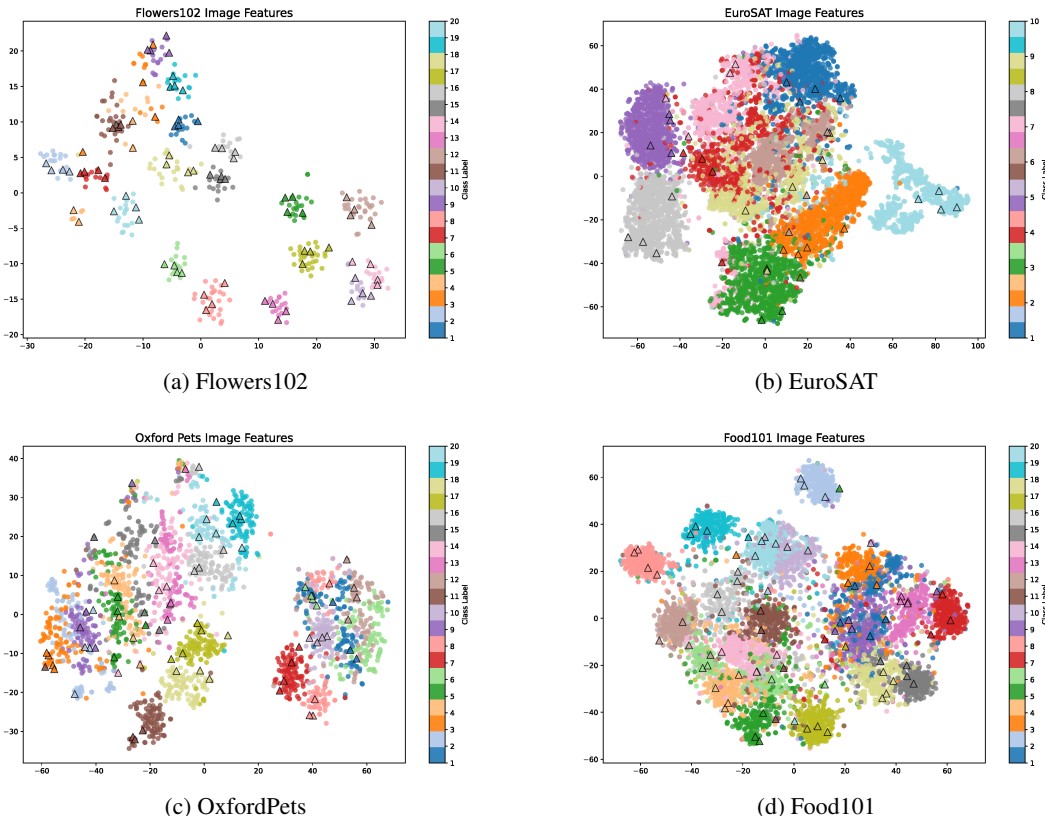

(a) Flowers102

(b) EuroSAT

(c) OxfordPets

(d) Food101

Figure 9: t-SNE visualization of image features from the CLIP ResNet50 model across four datasets: Flowers102, EuroSAT, Oxford Pets, and Food101. Circles represent test image samples, while triangles indicate few-shot samples. The visualization focuses on the first 20 categories and a 4-shot scenario for clarity.

