# OpenReview forum: "Efficient and Context-Aware Label Propagation for Zero-/Few-Shot Training-Free Adaptation of Vision-Language Model"
_ICLR.cc/2025/Conference — ICLR 2025 Poster_

### Official Review · Reviewer_9AGk · 2024-10-27

**Soundness:** 4
**Presentation:** 4
**Contribution:** 4
**Rating:** 8
**Confidence:** 5

**Summary:**

This paper proposes graph-based efficient adaption and inference for vision-language models. Specifically, the paper maximizes the use of test samples through iterative label propagation without the need for task-specific tuning. Additionally, the paper introduces a novel method to mitigate biases in VLMs through context-aware re-weighting and a interesting approach for identifying KNNs within each modality. Extensive experiments on various downstream tasks demonstrate the effectiveness of the proposed graph-based method.

**Strengths:**

1. The proposed use of K-nearest neighbors within each modality and context-aware edge re-weighting are novel approaches that align well with the property of pre-trained VLMs.
2. The paper is well-written and easy to understand.
3. Extensive experiments were fairly conducted with recent baselines across a range of datasets and architectures.
4. The paper improves the feasibility of the proposed method by using dynamic graph expansion, supported by a practical analysis of wall-clock time in Table 6 and time complexity in Appendix A.1.

**Weaknesses:**

I think the paper has no significant weaknesses.
1. Evaluating the performance degradation when using one-step label propagation instead of iterative label propagation could provide valuable insights for practitioners. This analysis would shed light on the trade-off between reduced complexity and potential accuracy decline, assisting in making informed decisions for applications where computational efficiency is crucial.
2. When utilizing an additional 16-shot training samples in Table 2, it appears that ECALP uses the training data as a component of the graph rather than for training. Wouldn’t it be possible to apply ECALP in addition to traditional prompt learning methods such as CoOp and CoCoOp?

**Questions:**

1. In Figure 1, it seems that the second image encoder and text encoder should be switched.
2. In Line 161, how can we obtain multiple textual prompts $z_{cm}$ for the $c$-th class? It seems that the value of $M$ can vary from class to class.

---

> ### Author Response · Authors · 2024-11-21
> **Response to Reviewer 9AGk**
>
> ## Response to Weaknesses
>
> 1. **Performance of One-step Label Propagation**
>
>     Thank you for the suggestion on evaluating one-step versus iterative label propagation. As detailed in **Appendix A.2.5**, ECALP demonstrates increased accuracy with more iterations, as shown by our results on the DTD dataset. While a single iteration achieves 53.31% accuracy, three iterations enhance this to 54.49%, balancing efficiency and performance. This analysis highlights the trade-off between computational simplicity and accuracy, aiding practitioners in making informed choices for applications where computational resources are limited.
>
> 2. **Combination with CoOp**
>
>     Thank you for your insightful question. In **Appendix A.2.4**, we explored integrating ECALP with traditional prompt learning methods like CoOp, using 1 to 16-shot samples as part of the graph rather than for training. Our results showed that combining ECALP with CoOp significantly enhances CoOp performance, demonstrating that ECALP can effectively complement traditional methods, further validating its applicability and versatility.
>
> ## Response to Questions
>
> 1. **Mistake in Figure 1**
>    We appreciate your careful review and apologize for the oversight. In **Fig. 1**, we have corrected the error by switching the positions of the second image encoder and the text encoder. Thank you for bringing this to our attention.
>
> 2. **Multiple Textual Prompts**
>
>    For each category, multiple textual prompts are generated using various templates, such as 'a photo of a {}', 'a sculpture of a {}', and 'a rendering of a {}'. These templates are suggested by the original CLIP paper. This method produces multiple text embeddings, which we then average to achieve a robust text representation. Crucially, we apply the same set of templates to all categories within a dataset, ensuring consistency across classes.
>
> We hope these additions address your concerns thoroughly and demonstrate the robustness and practicality of our approach. Thank you again for your feedback, which has significantly contributed to improving our manuscript.
>
> Kind regards,
> The Authors

---

> > ### Comment · Reviewer_9AGk · 2024-11-22
> >
> > Thank you for your detailed response. The combination results with CoOp are also interesting. I plan to retain the score.

---

> > > ### Author Response · Authors · 2024-11-22
> > >
> > > Thank you for recognizing our work. We appreciate your positive feedback.

---

### Official Review · Reviewer_uWXi · 2024-11-01

**Soundness:** 3
**Presentation:** 3
**Contribution:** 3
**Rating:** 6
**Confidence:** 4

**Summary:**

The paper proposes a approach for adapting vision-language models (VLMs) using efficient, context-aware label propagation. It introduces dynamic graph construction to improve inference accuracy without task-specific tuning, and employs context-aware feature re-weighting for better task adaptation. The method demonstrates superior performance and efficiency across various downstream tasks, highlighting its potential in zero-/few-shot scenarios.

**Strengths:**

1. The introduction of dynamic graph expansion is innovative, allowing for real-time adaptation and efficient use of test samples.
2. The method enhances feature relevance by re-weighting based on context, which is a novel approach to improving model adaptability to downstream tasks. This can potentially lead to better performance in diverse scenarios.
3. Employing an iterative solution rather than a closed-form one reduces computational costs and enhances scalability. This is particularly beneficial for handling large datasets.

**Weaknesses:**

1. More detailed motivation behind the model design is preferred. It is important to explain why the authors propose the method in this work.
2. The method may be complex to implement, particularly for large-scale or varied tasks. Detailed guidelines for implementation would be beneficial.
3. The approach assumes that the context-aware feature re-weighting will generalize well across different tasks, but this might not hold true for all types of data or in cases with significant domain shifts.

**Questions:**

Please refer to the weakness.

---

> ### Author Response · Authors · 2024-11-21
> **Response to Reviewer uWXi**
>
> ## Response to Weaknesses
>
> 1. **Motivation of ECALP**
>
>     Our method is motivated by the need to enhance VLMs without task-specific tuning, leveraging test samples more effectively. We propose a graph-based approach for label-efficient adaptation, dynamically constructing graphs over text prompts, few-shot examples, and test samples. This allows for inference without the need for additional support sets, improving feasibility and memory efficiency. Our model includes context-aware feature re-weighting to address modality differences and enhance task adaptation accuracy. By supporting efficient graph expansion and iterative label propagation, our method enables real-time inductive inference. Extensive evaluations demonstrate superior performance across various tasks, including fine-grained categorization, style-transfer classification, and out-of-distribution generalization.
>
> 2. **Method Implementation**
>
>     Thank you for your suggestion regarding implementation complexity. Our method is designed to be straightforward and does not require task-specific tuning. We have provided detailed implementation guidelines in **Section 3.5**, along with Algorithm 1. Additionally, we promise to release the source code upon acceptance, aiming to contribute to the community and foster the development of more advanced methods in the future.
>
>
>
> 3. **Generalization to Significant Domain Shifts**
>
>     We appreciate the concern regarding the generalization of context-aware feature re-weighting across different tasks. Our approach leverages robust text and few-shot re-weighting, which combines prompt text embeddings with target distribution information, minimizing the impact of domain shifts. As shown in **Appendix A.2.8**, we provide visualizations of severe corruption from the ImageNet-C dataset, illustrating our method's resilience. Furthermore, as detailed in **Appendix A.2.6**, ECALP consistently outperforms CLIP-ResNet50 by 3%-5% across all corruption severity levels, including minimal levels, demonstrating its robustness and effectiveness even under significant domain shifts.
>
> We hope these additions address your concerns thoroughly and demonstrate the robustness and practicality of our approach. Thank you again for your feedback, which has significantly contributed to improving our manuscript.
>
> Kind regards,
> The Authors

---

> > ### Comment · Reviewer_uWXi · 2024-11-25
> >
> > Thanks for the authors' responses. The authors have addressed my concerns and I will maintain my score.

---

> > > ### Author Response · Authors · 2024-11-25
> > >
> > > Thank you for recognizing our work. We appreciate your positive feedback.

---

### Official Review · Reviewer_qcpB · 2024-11-03

**Soundness:** 3
**Presentation:** 3
**Contribution:** 2
**Rating:** 6
**Confidence:** 3

**Summary:**

This paper proposes a graph-based approach for label-efficient adaptation and inference. The method dynamically constructs a graph based on available samples to enable real-time inductive inference.

**Strengths:**

1. The method requires no additional unlabeled support set and effectively leverages the test sample manifold through dynamic graph expansion.
2. This paper is clear and easy to follow.
3. This paper conducts extensive evaluations on diverse downstream tasks.

**Weaknesses:**

1. Compared to the SOTA methods, the performance improvement seems limited. Additionally, the second-highest performance can be highlighted with an underline for clarity.
2. In the computational efficiency section, does the testing time for ECALP include the dynamic graph construction process, or does it only account for the label propagation time?

**Questions:**

1. It appears that ECALP requires additional storage. How does its CUDA memory compare to that of previous methods?
2. If the initial testing samples are used to create an incorrect adjacency graph, will this lead to a cumulative error?
3. In corrupted downstream tasks, is ECALP still effective when the severity is low (e.g., at level 1)?
4. Is the method sensitive to the label propagation iterations T?

---

> ### Author Response · Authors · 2024-11-21
> **Response to Reviewer qcpB**
>
> ## Response to Weaknesses
>
> 1. **Limited Performance Improvement & Highlight Second-highest Performance**
>
>     We appreciate the reviewer's feedback on performance gains. Enhancing VLMs in a training-free setting is inherently challenging. Despite this, ECALP achieves a 1-2% accuracy improvement over state-of-the-art methods in tasks like fine-grained recognition, style transfer, and out-of-distribution scenarios. These improvements are notable given the absence of hyperparameter tuning, highlighting ECALP's robustness and practicality.
>
>    Additionally, we have underlined the second-highest performance in **Tab. 1, 2, and 3** for improved clarity. Thank you for this suggestion.
>
> 2. **Clarify the Computational Efficiency Measurement**
>
>     In **Tab. 6**, the testing time for ECALP includes the entire process: dynamic graph construction, label propagation, data loading, and CLIP encoder operations. This comprehensive measurement demonstrates our method's computational efficiency in practical scenarios.
>
>
> ## Response to Questions
>
> 1. **CUDA Memory Comparison**
>
>     ECALP is designed to be memory-efficient. On the DTD dataset, ECALP requires approximately 2.4 GB of CUDA memory per test image, compared to 2.3 GB for the CLIP ResNet 50 baseline and 3.0 GB for ZlaP. In contrast, training-required methods like TPT consume about 21.9 GB during training. Our efficiency is primarily due to: (1) being training-free; (2) storing only the image features as graph nodes, each occupying just 4 KB of CUDA memory; and (3) implementing label propagation with sparse operations, which conserves memory.
>
> 2. **Robustness to Incorrect Adjacency Graph Initialization**
>
>     To evaluate the robustness of our dynamically constructed graph for label propagation, we conducted experiments under various data-stream conditions, as detailed in **Appendix A.2.7**. We test ECALP's performance with three different test set configurations: Standard Random Sampling: A typical random sample of the test stream. Hard Samples First: The initial 5% of the test stream consists of only hard samples (those misclassified by CLIP baseline), followed by a random sampling for the remaining 95%. 10% Hard Samples First: The initial 10% of the test stream contains hard samples, with the rest being randomly sampled.
>
>     The results below suggest a minimal performance drop, with less than a 0.4% decrease in accuracy even with a 10% hard sample initialization. This resilience is due to our graph's dynamic nature, where edge connections between data points are continuously updated as the test sequence progresses. This adaptability ensures that our method remains robust against poor initializations, preventing cumulative errors and maintaining high accuracy even under challenging conditions.
>
> | Initialization Method      | Accuracy |
> |----------------------------|----------|
> | Random Initialization      | 54.49    |
> | 5% Hard Samples First     | 54.31    |
> | 10% Hard Samples First    | 54.14    |
>
> 3. **Performance on Different Corruption Severities**
>
>     ECALP remains effective at different severity levels in corrupted downstream tasks. As detailed in **Appendix A.2.6**, we examined ECALP's performance using the ImageNet-C dataset across severity levels from 1 to 5. ECALP consistently outperforms CLIP-ResNet50 by approximately 3%-5% across all severity levels. This demonstrates ECALP's robustness and effectiveness regardless of the corruption severity.
>
> 4. **Impact of Label Propagation Iterations**
>
>     The performance of ECALP is related to the number of label propagation iterations \( T \). As detailed in **Appendix A.2.5**, we explored varying T  from 1 to 6 on the DTD dataset. The results suggest that slightly more iterations may improve the accuracy. This is due to a larger receptive field by more propagation iterations, which better utilize the test data's manifold structure. However, increasing T also increases computational costs, and we chose T = 3 to strike a balance between efficiency and accuracy.
>
> The table below shows the accuracy at different iterations:
>
> | Iterations ( T ) | Accuracy (%) |
> |----------------------|--------------|
> | 1                    | 53.31        |
> | 2                    | 54.37        |
> | 3                    | 54.49        |
> | 4                    | 54.73        |
> | 5                    | 54.96        |
> | 6                    | 54.96        |
>
> We hope these additions address your concerns thoroughly and demonstrate the robustness and practicality of our approach. Thank you again for your feedback, which has significantly contributed to improving our manuscript.
>
> Kind regards,
> The Authors

---

> > ### Comment · Reviewer_qcpB · 2024-11-23
> >
> > Thank you for your response. I will keep the positive score.

---

> > > ### Author Response · Authors · 2024-11-23
> > >
> > > Thank you for recognizing our work. We appreciate your positive feedback.

---

### Official Review · Reviewer_9D8M · 2024-11-03

**Soundness:** 4
**Presentation:** 4
**Contribution:** 3
**Rating:** 6
**Confidence:** 3

**Summary:**

The paper proposes ECALP, a novel graph-based framework for training-free adaptation of vision-language models. The proposed method constructs a graph over text prototypes, optional training samples and testing samples in a simple, dynamic fashion, and adopts label propagation for inference. Furthermore, edge re-weighting is carefully designed to take into account the semantic context of different classes. Throughout training and inference, the framework introduces little augmentation and hyper-parameter search, showing its efficiency and practicality. Finally, comprehensive experimental results over a wide range of datasets on zero-shot/few-shot fine-grained classification, style-transfer and out-of-distribution detection tasks show significant improvement in performance of the proposed approach over previous methods.

**Strengths:**

1. Comprehensive experiments and great results:

The paper provides a comprehensive comparison of ECALP against a number of highly competitive prior works, and carried out experiments on a variety of datasets as well as visual tasks, which is sufficient to validate the approach. The empirical results clearly demonstrate that ECALP achieves significantly better performance than previous methods. The experimental design, including the main setting, comparison baselines as well as ablation studies, are well justified. Moreover, the method proves to be computationally efficient.

2. Nice presentation and clear structure

The figures in this paper are well-designed and effectively highlight the pipeline as well as advantages of the proposed framework. The formulation, presentation of the results as well as the visuals are aesthetically pleasing and easy to follow.

3. Clear motivation and good writing

The introduction and abstract are well written and lays a good foundation for the paper. It is easy to see the motivation behind the work.

**Weaknesses:**

1. In Figure 1, the inputs to the second image encoder and the text encoder seem to be mixed up.

2. As is discussed in the paper, ECALP builds upon similar ideas with ZLaP (i.e. label propagation), and although I acknowledge that the proposed method achieves improvement in performance and eliminates the need for additional unlabelled datasets, the contribution seems somewhat limited.

3. In the detailed few-shot results provided in the appendix, it seems that ECALP does not always achieve the best performance. Especially on datasets like OxfordPets and Food101, the accuracy of ECALP even goes down as the number of samples per class increases. This weakens the claim of ECALP’s robustness in low-shot scenarios.

**Questions:**

1. As is mentioned above, I wonder if the authors could offer some explanation on ECALP’s behavior on certain datasets in low-shot settings.

2. In the ablation study, in Table 4, it’s interesting to see that without Text Reweight, the model achieves slightly better performance under the 16-shot setting on ImageNet. It would be nice if the authors could shed some light on this phenomenon.

---

> ### Author Response · Authors · 2024-11-21
> **Response to Reviewer 9D8M**
>
> ## Response to Weaknesses
>
> 1. **Mistake in Figure 1**
>    We appreciate your careful review and apologize for the oversight. In **Fig. 1**, we have corrected the error by swapping the positions of the second image encoder and the text encoder. Thank you for bringing this to our attention.
>
> 2. **Contribution Relative to ZLaP**
>    While ECALP shares a similar foundational idea with ZLaP, specifically in leveraging data manifolds via label propagation, our approach introduces several novel contributions. These include i) reduced data requirements, ii) supporting few-shot adaptation, iii) context-aware edge re-weighting, iv) improved label propagation strategy, and v) improved performance. Notably, our zero-shot ECALP does not require an additional support set for graph construction, enhancing feasibility and memory efficiency in real-world applications. In contrast, a separate support set is required by ZLaP.  ECALP also supports **Few-shot Adaptation** by incorporating few-shot samples into the graph with reweighting, unlike ZLaP, which is limited to zero-shot adaptation. To address modality differences and reduce irrelevant information impact, we introduce a **Context-aware Edge Re-weighting**. Our approach enhances label propagation efficiency through **Iterative Label Propagation** and **Dynamic Graph Expansion**, enabling low-cost inductive inference. We demonstrate superior performance across diverse tasks, including fine-grained datasets, style-transfer datasets, and OOD datasets.
>
> 3. **Few-shot Performance on OxfordPets and Food101**
>     We appreciate the observation regarding performance variations on the Oxford Pets and Food101 datasets. We analyze the causes for the relatively poorer few-shot results on these datasets through a qualitative visualization of testing data features in **Appendix A.4**. Two datasets, Flowers102 and EuroSAT, are brought in for comparative studies. The testing sample features are projected into 2D space via t-SNE. We make the following observations. The testing samples are more cluttered and mixed up in OxfordPets and Food101 than in Flowers102 and EuroSAT. There are more separated sub-classes, i.e. isolated clusters within each semantic class, for OxfordPets and Food101. This suggests having a few-shot labeled samples is less effective for OxfordPets and Food101 datasets. In certain cases, an inappropriate selection of few-shot samples may even bias the adaptation.
>
> ## Response to Questions
>
> 1. **Analysis of Text Reweighting on ImageNet**
>    In zero-shot scenarios, text re-weighting effectively bridges modality differences between text and image embeddings by mitigating the impact of irrelevant features. However, in few-shot adaptation scenarios, feature selection is mainly carried out by the few-shot samples which share the same representation space with other testing samples. This explains why improvement is mainly brought in by few-shot reweighting, rather than text reweighting. An insignificant drop of performance on ImageNet might be caused by the conflict between few-shot and text reweighting. Nevertheless, they are complementary to each other on DTD and UCF datasets.
>
> We hope these clarifications comprehensively address your concerns and underscore the robustness and applicability of our approach. We are grateful for your insightful feedback, which has greatly contributed to enhancing our manuscript.
>
> Kind regards,
> The Authors

---

> > ### Comment · Reviewer_9D8M · 2024-11-21
> > **Thank you for your detailed response.**
> >
> > Thank you to the authors for your detailed response. I appreciate the additional analysis on few-shot performance of ECALP, as well as the explanation of its contribution in comparison with ZLap, which effectively addressed most of my concerns.
> >
> > Overall, great rebuttal. I will maintain my original positive overall rating, and add one point to the soundness rating.

---

> > > ### Author Response · Authors · 2024-11-22
> > >
> > > Thank you for your thoughtful review and positive feedback. We're pleased that our clarifications have effectively addressed your concerns.

---

### Author Response · Authors · 2024-11-21
**General Response to Reviewers' Comments on Manuscript Revision**

We appreciate the constructive feedback provided by the reviewers, which has significantly helped us enhance the clarity and depth of our manuscript. Below, we outline the specific revisions made in response to each comment:

1. **Combination with CoOp**

    In **Appendix A.2.4**, we explored integrating our method, ECALP, with the traditional prompt learning approach CoOp. This combination was tested with varying numbers of training samples from 1 to 16 shots. Our results showed that CoOp combined with ECALP further enhances CoOp's performance significantly. This demonstrates that ECALP can be seamlessly integrated with traditional prompt learning methods to boost their performance, further validating our applications.

2. **Impact of Label Propagation Iterations**

    In **Appendix A.2.5**, we investigated how varying the number of label propagation iterations affects performance. Increasing iterations generally improved accuracy by better utilizing the data manifold but also increased computational demands. We found an optimal balance by selecting a moderate number of iterations to maintain efficiency without sacrificing much accuracy. This illustrates the trade-off between computational cost and performance enhancement. Moreover, multiple iterations of label propagation show significant advantages over a single iteration, indicating the effectiveness of our propagation approach.

3. **Performance on Different Corruption Severities**

    In **Appendix A.2.6**, we tested our method on tasks with various levels of data corruption. ECALP consistently outperformed the CLIP baseline by 3% to 5%, demonstrating robust handling of corrupted data. This robustness is crucial for real-world applications where data quality may vary significantly.

4. **Robustness to Bad Initialization**

    In **Appendix A.2.7**, we evaluated how well our approach handles poor initial conditions during testing. By dynamically updating relationships between data points, our method shows minimal performance drop even when starting with challenging data samples. This robustness is vital for deploying the method across diverse and unpredictable real-world tasks.

5. **Visualization Samples for Corruption Task**

    In **Appendix A.2.8**, we provide visualizations of sample images subjected to various types of corruption from the ImageNet-C dataset. These visualizations are intended to illustrate the types of distortions and challenges posed by severe corruption conditions.

6. **Correction in Figure 1**

    In **Fig. 1**, we corrected a mistake by swapping the positions of the second image encoder and text encoder. We thank the reviewers for pointing this out.

7. **Highlighting Second-highest Performance**

    In **Tab. 1, 2, and 3**, we have underlined the second-highest performance for clarity. Thanks for this suggestion.

---

### Meta-Review · Area_Chair_Cc4H · 2024-12-19

**Metareview:**

This paper proposes a novel framework, called ECALP, an approach designed to efficiently adapt vision-language models (VLMs) through the construction of a graph over text prototypes. ECALP can dynamically construct a graph over text prompts, few-shot examples, and test samples, using label propagation for inference without task-specific tuning. The experiments of this work is extensive and solid, with remarkable performance on several downstream tasks. Overall, I found the methodological design novel and the experimental results solid, and therefore would like to suggest accept this work.

**Additional Comments On Reviewer Discussion:**

There are some minor concerns raised, primarily regarding the limited performance improvement in specific tasks (e.g., few-shot scenarios) and the computational overhead compared to state-of-the-art methods. In their rebuttal, the authors effectively addressed these concerns and discussed the potential enhancements of their approach in areas such as fine-grained recognition, style transfer, and out-of-distribution scenarios. The reviewers are all positive toward the rebuttal.

---

### Decision · Program_Chairs · 2025-01-22

Accept (Poster)